# MicroRNAs in Learning and Memory and Their Impact on Alzheimer’s Disease

**DOI:** 10.3390/biomedicines10081856

**Published:** 2022-08-01

**Authors:** I-Fang Wang, Pei-Chuan Ho, Kuen-Jer Tsai

**Affiliations:** 1Ph.D. Program for Neural Regenerative Medicine, College of Medical Science and Technology, Taipei Medical University, Taipei 11031, Taiwan; ifwang@tmu.edu.tw; 2Ph.D. Program in Medical Neuroscience, College of Medical Science and Technology, Taipei Medical University, Taipei 11031, Taiwan; 3Institute of Clinical Medicine, College of Medicine, National Cheng Kung University, Tainan 70403, Taiwan; peggy821124@gmail.com; 4Research Center of Clinical Medicine, National Cheng Kung University Hospital, College of Medicine, National Cheng Kung University, Tainan 70403, Taiwan

**Keywords:** Aβ pathology, Alzheimer’s disease, learning/memory impairment, microRNA

## Abstract

Learning and memory formation rely on the precise spatiotemporal regulation of gene expression, such as microRNA (miRNA)-associated silencing, to fine-tune gene expression for the induction and maintenance of synaptic plasticity. Much progress has been made in presenting direct evidence of miRNA regulation in learning and memory. Here, we summarize studies that have manipulated miRNA expression using various approaches in rodents, with changes in cognitive performance. Some of these are involved in well-known mechanisms, such as the CREB-dependent signaling pathway, and some of their roles are in fear- and stress-related disorders, particularly cognitive impairment. We also summarize extensive studies on miRNAs correlated with pathogenic tau and amyloid-β that drive the processes of Alzheimer’s disease (AD). Although altered miRNA profiles in human patients with AD and in mouse models have been well studied, little is known about their clinical applications and therapeutics. Studies on miRNAs as biomarkers still show inconsistencies, and more challenges need to be confronted in standardizing blood-based biomarkers for use in AD.

## 1. Introduction

### 1.1. MicroRNAs: Biogenesis/Functional Mechanism in Neurons

MicroRNAs (miRNAs) are small non-coding RNAs (~19–24 nucleotides) that primarily act as post-transcriptional regulators of approximately 60% of protein-coding genes [1]. By incorporation into the RNA-induced silencing complex, miRNAs serve as guides via sequence-specific, partial base pairing with their recognition sites in the 3 untranslated regions of the target mRNAs [2]. As fine regulators of gene expression that participate in many pivotal biological processes, miRNAs play important roles in a range of signaling pathways in post-mitotic neurons that undergo extensive dynamic changes and modifications for a huge network of connectivity; they mediate activity-dependent local protein translations and influence dendritic spine morphology (size and density), spine formation, and maturation [3]. Furthermore, miRNAs play important roles in complex behavioral phenomena such as long-lasting forms of synaptic plasticity that underlie memory formation, retrieval, and consolidation [4,5].

### 1.2. Learning and Memory

The formation of new memories is a complex process that requires: (1) activity-dependent gene transcription; (2) newly synthesized proteins, and (3) fine-tuned neuronal networks, such as memory traces/engrams, to strengthen synaptic connections and sustain neuronal plasticity [6]. The ability for spatial learning and memory formation allows most animal species to progressively adjust their behavior to adapt to complex environments, which is critical for their survival. This can be assessed in laboratory animals by applying behavioral paradigms such as the Morris water maze (MWM) and Barnes maze [7,8]. Nevertheless, to avoid detrimental situations, animals develop fear responses that help them to predict danger in seemingly neutral contexts. Contextual fear learning requires the coordinated activity of the hippocampus and amygdala; such behavioral tasks are mainly used in contextual fear conditioning (CFC) [9]. MWM and CFC are very stressful approaches, and positive or negative reinforcement as well as long training schedules are their disadvantages. The novel object recognition (NOR) test, which simply relies on the innate preference of mice for novelty, is also a commonly used behavioral assay for various aspects of learning [10].

Experimentally, inbred rodents have often been used to study the molecular and cellular basis of average phenotypes/behaviors to minimize the effects of genetic differences among tested individuals [11]. In other words, only the major phenotypes, on average, have been shown most of the time. However, between-mouse variations in tissue genes have been observed in isogenic laboratory mice [12]. Notably, genetically identical inbred rodents have been shown to exhibit various behavioral traits [13]. Whether variations in tissue gene expression among genetically identical animals play a causative role in the environmentally induced variation of phenotypes/behaviors remains to be fully elucidated. Previously, we demonstrated that the underlying mechanisms of behavioral variation are correlated with a specific miRNA cluster. In particular, we showed that the stochastic activation of miR-466f-3p through CREB activation modulates individual variations in spatial learning and memory capability in inbred mice [5], suggesting a functional role of miRNAs in regulating learning-derived behavioral variability.

Alzheimer’s disease (AD) is the most common cause of dementia among elderly people. It is a progressive disease with dementia symptoms that gradually worsen over time, from memory loss and changes in behavioral and emotional responses to an inability to respond to their environment [14]. Extracellular plaque deposits of the β-amyloid peptide (Aβ) and neurofibrillary tangles (NFT) composed of the microtubule-associated protein (MAP) tau are the two major pathological hallmarks of AD. Aβ aggregation is the causative factor for AD development, as it induces significant cytotoxicity in neurons, leading to synaptic damage, tau protein phosphorylation, inflammation, oxidative stress, apoptosis, and eventually, neuronal damage and death [15]. In addition to Aβ, there is an accumulation of intracellular insoluble hyperphosphorylated tau in AD. Tau is the most abundantly expressed MAP in the neurons. Abnormally phosphorylated tau shows a reduced binding affinity for microtubules, leading to oligomeric tau aggregation and NFT formation [16].

The fundamental aspects of miRNAs in neurobiology have been extensively studied over the past 20 years, and there are immense possible clinical implications for miRNA’s function in many fields of neuroscience. Much progress has been made in understanding the role of miRNAs, including the use of various approaches from classical genetic and virus-mediated overexpression to inhibition via “sponges” or “antagomiRs”. In the following sections, we discuss the known functions and dysfunctions of several miRNAs in regulating learning and memory in AD and highlight their potential use as biomarkers in the diagnosis and therapeutics of cognitive disorders.

## 2. Role of miRNAs in Learning, Memory, and Cognitive Disorders

### 2.1. Direct Evidence of miRNA Regulation in Learning and Memory

Brain-specific or brain-enriched miRNAs, some of which are transcriptionally induced or promote the turnover of mature forms by neuronal activity, are widely expressed in different brain regions [17]. It is now well established that the biogenesis, activity, and degradation of specific miRNAs regulate neuronal plasticity, which is responsible for many complex brain functions, including the learning/memory process, and that the misexpression of some miRNAs is associated with neurological disorders [18,19]. For example, the overexpression of miR-34a in neurons was found to negatively affect dendritic growth and arborization and weakened synaptic plasticity by reducing synaptotagmin-1 expression in cortical neuronal cultures [20]. In animal experiments, auditory fear conditioning upregulated miR-34a in the basolateral amygdala, whereas antagonizing miR-34a by miRNA sponges suppressed auditory fear memory [21]. Virus-mediated overexpression of miR-34a in the lateral ventricle enhanced MWM performance by promoting neural progenitor proliferation [22]. However, transgenic overexpression of miR-34a in the whole brain exhibited profound behavioral impairment in the T-maze task, accompanied by accumulation of intracellular Aβ and tau hyperphosphorylation [23]. Overexpression of miR-34c by the injection of mimics or lentivirus into the hippocampus of wild-type (WT) mouse brain impaired learning and memory formation in multiple behavioral experiments, including MWM, CFC, and NOR. However, the expression of miR-34c was shown to increase with age in mice and in humans with the onset of AD, contributing to impairments in CFC memory. When miR-34c inhibitors were injected into the hippocampus, the memory-impairing phenotypes were rescued in an aged AD mouse model (APP/PS1-21) by targeting Sirtuin 1 (SIRT1) [24]. Similar results were observed in another AD mouse model (SAMP8), indicating that miR-34c mediates synaptic and memory deficits (tested using the MWM) by targeting SYT1 in the ROS-JNK-p53 pathway [25].

#### 2.1.1. miRNAs Involved in CREB-Dependent Transcription and CREB-Regulated miRNAs

In the invertebrate *Aplysia californica*, miR-124 is exclusively enriched in sensory neurons, both soma and processes, and regulates serotonin-induced synaptic plasticity via CREB regulation [26]. miR-124 is an abundant, brain-specific miRNA. Reducing miR-124 levels through the injection of an locked nucleic acids (LNA)-probe was found to restore spatial memory and social interaction in adult mice carrying an EPAC-null mutation, whereas AAV-mediated overexpression of miR-124 in the hippocampus impaired long-term potentiation and spatial memory in EPAC^+/+^ mice by regulating the expression of Zif268 [27], demonstrating the prominent role of miR-124 in the negative regulation of synaptic plasticity and memory formation.

The NAD-dependent deacetylase SIRT1 is essential for normal associative learning in the CFC task, and this function requires crosstalk with the brain-specific miR-134. The transcriptional regulation of miR-134 is repressed by SIRT1, which cooperates with the YY1 DNA-binding element, and miR-134 expression is upregulated in SIRT1-deficient mice. Similar to miR-124, the overexpression of miR-134 negatively regulates fear memory formation and long-term potentiation induction in the rodent hippocampus through translational repression of CREB mRNA; therefore, it affects learning and memory in the CFC paradigm by mediating the CREB-BDNF-dependent signaling pathway [28].

In addition to being a miRNA target, CREB also regulates miRNA transcription. Being co-transcribed from a single locus containing a functional CRE in the promoter, the miR-132/miR-212 cluster can simultaneously be induced by neuronal activity-dependent modulation of CREB [29]. Predictably, miR-132 manipulation was accompanied by changes in memory performance. Lentivirus-mediated interference of miR-132 in the hippocampus or double-knockout miR-132/212 in the forebrain impaired trace fear conditioning, NOR, CFC, and performance in the Barnes maze in mice [29,30], whereas the overexpression of miR-132 in the hippocampus or perirhinal cortex impaired NOR memory in mice [31] and rats [32]. Hansen et al., introduced an inducible miR-132 transgene in the hippocampus of a mouse strain and demonstrated that the expression of relatively low levels of transgenic miR-132 (1.5-fold), which is similar to the physiological induction of miR-132 in spatial memory tasks, could significantly enhance cognitive capacity. In contrast, the overexpression of high levels (3-fold) of miR-132 inhibited learning [33]. This finding indicates that miRNA expression must be maintained within a limited range to ensure normal functioning. Similar to miR-132, we previously found that neuronal activity induces miR-466f-3p through the transcriptional activation of CREB [5]. However, unlike miR-132, which is stress-inducible, miR-466f-3p is only induced in mice with good performance on the MWM task [34]. miR-466f-3p appears to be a positive regulator of neuronal plasticity via the CREB → pCREB → miR-466f-3p → MEF2A axis during spatial learning and memory formation.

#### 2.1.2. Regulation of Fear Consolidation and Extinction by miRNAs—From Exposure to Inhibitory Learning

Recent examples from the literature further lengthen the list of specific miRNAs thought to be involved in several types of learning-related behaviors and memory formation processes in the mammalian brain, some of which have opposite effects. Several studies have demonstrated a role of miRNAs in amygdala-dependent fear learning. For example, miR-182 was found to be downregulated in the mouse lateral amygdala in vivo after auditory fear conditioning. The overexpression of miR-182 in the lateral amygdala disrupted long-term fear memory [35]. Conversely, miR-182/96/183, which belong to the same miRNA cluster, were induced in the mouse hippocampus during the NOR task training. Mimicking this increase by the overexpression of miR-183/96/182 enhanced object memory, whereas the knockdown of endogenous miR-183/96/182 impaired it. This effect involved the modulation of several neuronal-plasticity-related genes such as *HDAC9* [36]. In contrast to miR-182, elevated miR-126a-3p levels contributed to the consolidation of contextual fear memory by modulating its target (EFHD2) in WT mice. Decreasing miR-126a-3p using antagomiR impaired the consolidation of CFC, spatial memory (MWM), and recognition memory (NOR) but not cued fear memory, whereas the overexpression of miR-126a-3p in the dentate gyrus of the hippocampus reduced the Aβ plaque area and neuroinflammation as well as rescued contextual fear memory deficits in the APP/PS1 AD mouse model [37]. The overexpression of miR-135b-3p in the basolateral amygdala not only enhanced remote fear memory in stress-resilient mice but also in the serum of military veterans suffering from post-traumatic stress disorder (PTSD), indicating that miRNAs play roles in fear- and stress-related disorders [38]. In contrast to miR-134 and miR-126a-3p, miR-128b levels in the basolateral amygdala increased only after fear extinction, a learned safety-related fear inhibitory paradigm. The knockdown of miR-128b impaired the formation of fear extinction memory, whereas the forced expression of this miRNA in the mouse infralimbic prefrontal cortex facilitated fear extinction, indicating that miR-128b is specific to this form of inhibitory learning by suppressing genes such as *Reelin*, *Creb1*, and *Rcs* [39]. More specifically, learned safety is a fear inhibitory mechanism that has potential as an experimental model for PTSD and depression. The role of miR-132/-212 in stress-associated, amygdala-dependent learning safety has also been clearly demonstrated [40], suggesting that miRNAs are involved in inhibitory emotional learning and memory. Table 1 and Table 2 summarize studies that have reported the manipulation of miRNA levels in either WT rodents or disease models with changes in cognitive performance.

### 2.2. Direct Evidence of miRNA Involvement in Cognitive Impairment

#### 2.2.1. miRNA and AD

There is substantial evidence that miRNAs are involved in the pathophysiology of the age-associated forms of dementia and neurodegeneration. AD is the most common type of dementia in the elderly, and extensive studies on AD have revealed altered miRNA expression profiles. Many miRNAs are severely dysregulated in the brains of AD patients, including miR-34c (upregulated [24]), miR-124-3p (downregulated [72,73]; upregulated [51]), miR-126a-3p (downregulated [72,74]), miR-128b (upregulated [53]), miR-132/-212 (downregulated [74,75,76]), miR-146a (upregulated [72,77]), miR-188-5p (downregulated [61]), miR-195 (downregulated [64]), miR-206 (upregulated [67]), miR-338-5p (downregulated [68]), and miR-485-5p (downregulated [75]) vs. miR-485-3p (upregulated [71]). Figure 1 shows growing evidence that miRNA dysregulation correlates with some major and important aspects of AD pathology that have been proven to cause cognitive impairment in animal models.

#### 2.2.2. miRNAs Are Involved in Aβ Production and Metabolism

Increasing evidence suggests that miRNAs play a role in regulating Aβ production/metabolism; therefore, Aβ-targeted miRNAs may have therapeutic implications for AD. Aβ peptides are produced from amyloid precursor proteins (APP) after cleavage by the β-site APP cleavage enzyme 1 (BACE1). Several miRNAs, including miR-31-5p, miR-126a-3p, miR-135b, miR-188-3p, miR-195, and miR-338-5p, participate in APP lysis by modulating BACE1 [78,79]. All these miRNAs were found to be downregulated in samples from AD patients or transgenic mice, showing a negative correlation with BACE1, which is highly expressed in the brains of AD patients. The overexpression of hippocampal miR-188-3p reduced BACE1 expression levels and Aβ formation and suppressed neuroinflammation in 5xFAD transgenic mice [62]. The overexpression of miR-126a has been found to be neuroprotective against Aβ42 toxicity, as discussed above [37]. In addition, miR-338-5p, a new miRNA that also targets BACE1, was significantly downregulated in the hippocampus of AD patients and in two animal models, namely 5xFAD and APP/PS1 transgenic mice. The overexpression of miR-338-5p in the hippocampus rescued spatial memory deficits in transgenic mice [68,69]. miR-338-5p is also associated with neuronal differentiation, neurogenesis, and neuronal protective effects through the negative regulation of BCL2L11, which attenuates amyloid plaque deposition, neuroinflammation, and neuronal apoptosis [68,69]. However, there are other miRNAs that increase Aβ levels. For example, miR-128 is involved in the development and progression of AD. In cell culturing, the inhibition of miR-128 attenuated Aβ-mediated cytotoxicity through the inactivation of the NF-κB pathway [80]. The levels of miR-128 and Aβ were significantly increased in the cerebral cortex of 3xTg-AD mice, whereas their target peroxisome proliferator-activated receptor gamma (PPARγ) was downregulated. The knockout of miR-128 attenuated an AD-like performance and alleviated cognitive deficits in 3xTg-AD mice by suppressing amyloid plaque formation, Aβ generation, and neuroinflammation by targeting PPARγ. In addition to animal models, miR-128 has been shown to be upregulated in the brain and plasma samples of AD patients [53,80]. These findings suggest that miR-128 is a useful biomarker for the inflammatory pathophysiology of AD.

#### 2.2.3. miRNAs Contribute to Abnormal Tau Protein Functions

miRNAs are closely related to the phosphorylation and pathological aggregation of tau proteins. Indeed, as a direct target of miR-132, the knockout of miR-132/212 leads to increased tau expression, phosphorylation, and aggregation. Conversely, the treatment of AD mice with miR-132 partially mimics a restored memory function and tau metabolism [56]. Additionally, the downregulation of miR-132/-212 in the hippocampus and prefrontal cortex of AD patients correlated with neuronal tau hyperphosphorylation, further elucidating the role of miR-132 in tauopathies [75]. Similarly, the expression of miR-132 and miR-212 in neural-derived extracellular vesicles has been shown to be decreased in AD patients [81]. Further, miRNAs can affect tau phosphorylation by regulating the activities of relevant enzymes. The overexpression of miR-125b leads to the upregulation of tau kinases, including p35, CDK5, and p44/42 MAPK (Erk1/2), whereas tau phosphatases (DUSP6, PPP1CA) and the anti-apoptotic factor Bcl-W were downregulated, all of which contribute to tau hyperphosphorylation in primary neurons. Moreover, injection of the miR-125b mimic into the hippocampus of WT mice impaired associative learning in the fear conditioning paradigm and was also accompanied by the downregulation of Bcl-W, DUSP6, and PPP1CA, resulting in increased tau phosphorylation in vivo [43]. The selective knockdown of miR-146a, the most commonly deregulated miRNA in developmental brain disorders, in the hippocampus of adult mice was found to cause severe learning and memory impairments, associated with a reduction in adult hippocampal neurogenesis, indicating for the first time a role for miR-146a in postnatal brain functions [46]. miR-146a was also highly expressed in the brains of AD patients and 5xFAD transgenic mice, and it promoted pathogenesis by modulating the ROCK1/PTEN signaling pathway, resulting in abnormal tau hyperphosphorylation in early NFT. The inhibition of miR-146a expression in the hippocampus resulted in enhanced hippocampal levels of ROCK1, repressed tau hyperphosphorylation, and a partly restored memory function in 5xFAD transgenic mice [59].

#### 2.2.4. miRNAs Mediate Synaptic Dysfunction

Synapse formation is the basis of neural signal transduction, whereas synaptic plasticity forms the basis of learning and memory. Memory impairment in AD patients results from abnormal synaptic plasticity [16]. Synapses are vulnerable to Aβ-induced neurotoxicity, and the dysregulation of some miRNAs may contribute to defective synaptic elimination and cognition in AD by inducing Aβ-mediated synaptic toxicity.

Kao et al., revealed the role of miR-34c in disrupting dendrites in primary hippocampal neurons [41]. Additionally, miR-34c is upregulated during Aβ accumulation by targeting the *VAMP2* gene. As miR-34c blockade upregulated VAMP2 expression levels and rescued Aβ-induced synaptic failure, learning and memory deficits were ameliorated [82]. Another study indicated that miR-124 levels were increased in the temporal cortex and hippocampus of AD patients and in a Tg2576 AD mouse model [51]. Induced levels of miR-124 recapitulated AD-like phenotypes in mice, including memory impairment and deficits in their synaptic transmission and plasticity, by directly regulating the expression of the target gene *PTPN1*. Thus, maintaining the balance of miR-124/PTPN1 levels by suppressing miR-124 can restore synaptic failure and memory deficits. These findings indicate that the miR-124/PTPN1 pathway is a critical mediator of synaptic dysfunction and memory loss in AD. In contrast, some miRNAs were found to be abnormally reduced in AD models and could have positive effects on neurons. miR-188-5p expression was found to be downregulated in the cerebral cortices and hippocampus of AD patients as well as in the brains of 5xFAD transgenic mice. The replenishment of miR-188-5p rescued Aβ-mediated synapse elimination and synaptic dysfunction, as well as impaired cognitive function by targeting NRP-2 in 5xFAD transgenic mice [61]. One of the targets of miR-132 is C1q, a classical complement cascade protein that mediates synapse elimination in the central nervous system and is highly expressed in AD patients. APP/PS1 transgenic mice transfected with miR-132 showed a significant increase in synaptic protein (PSD95, Synapsin-1, p-Synapsin) expression compared with the non-transfected AD group. These results suggest that miR-132 maintains synaptic plasticity by regulating C1q expression in AD [54].

#### 2.2.5. miRNAs Participate in Multiple Functions

In contrast to the above observations, several studies have reported a reduced expression of miR-124-3p in the neocortex and hippocampus of human AD patients compared with age-matched controls [72,73,75] as well as in an animal model of AD (APP/PS1). The downregulation of miR-124 expression and upregulation of complement C1q-like protein 3 (C1ql3) in the hippocampus and cerebral cortex resulted in Aβ deposition, cerebromicrovascular impairments, breakdown of the blood–brain barrier, and the promotion of angiogenesis [52]. miR-132 was also found to be downregulated in AD. A rat model of AD, established by the intracerebroventricular administration of Aβ, showed significantly decreased learning abilities, increased oxidative stress and cell apoptosis rates, and decreased levels of miR-132. Virus-mediated overexpression of miR-132 inhibited hippocampal iNOS expression and oxidative stress by regulating MAPK1 expression, thus improving spatial memory in the AD model [55].

Several studies have shown that miR-195 is involved in dementia-related pathological processes by directly targeting multiple genes (*APP*, *BACE1*, [47] *Cdk5r1*, [83] *Ppme1*, [84] *DR6*, [85], and *SYNJ1* [64]). While the upregulation of miR-195 in the hippocampus restored cognitive function in a chronic brain hypoperfusion model [47], ApoE4^+/+^KI mice [64], and APP/PS1 transgenic mice [63], the knockdown of endogenous miR-195 by the injection of antogomiR induced dementia in WT rats [47], thereby mimicking the cognitive deficits observed in AD pathology. The reduction in miR-195 levels has been further validated in a human brain parietal cortex and cerebrospinal fluid (CSF) samples, which are associated with the ApoE4^+/−^ genotype, disease progression (presented by clinical dementia rating scores and phospho-tau levels), and cognitive decline (measured by mini-mental state examination [MMSE]) during early AD development. The overexpression of miR-195 in ApoE4^+/+^KI mice with and without an AD background rescued cognitive deficits and pathological changes by regulating the ApoE-synj1-PIP2 pathway [64]. These findings strongly suggest that miR-195 is an important contributor to dementia and may be a potential clinical marker for the disease.

Moreover, both miR-485-3p and miR-485-5p have been reported to participate in AD pathology, but they have contrary expressions and effects on AD pathogenesis. miR-485-5p expression was downregulated in the hippocampus of APP/PS1 transgenic mice. The overexpression of miR-485-5p facilitated learning and memory capabilities (according to MWM and CFC tests), promoted pericyte viability, and suppressed Aβ40-induced pericyte apoptosis in APP/PS1 transgenic mice. Mechanistically, miR-485-5p directly targeted PACS1, a hallmark of AD, in pericytes [70]. As miR-485-5p was downregulated in the prefrontal cortex of patients with late-onset AD [75], miR-485-3p was overexpressed in the brain tissues, CSF, and plasma of patients with AD [71]. Furthermore, the knockdown of miR-485-3p enhanced Aβ clearance via CD36-mediated phagocytosis, decreased truncated tau levels, and reduced the secretion of proinflammatory cytokines, including interleukin-1β and tumor necrosis factor-α, eventually alleviating cognitive decline in 5xFAD transgenic mice. Taken together, these data suggest that miR-34c, miR-128, miR-146a, miR-485-3p, and other elevated miRNAs may be important mechanistic links in AD progression. Table 3 reviews the existing evidence regarding miRNAs involved in AD pathology.

#### 2.2.6. miRNA and Other Neurodegenerative Disorders

In addition to AD, other age-related dementias, including frontotemporal dementia (FTD), have been reported to be associated with deregulated miRNAs. In both patients with behavioral variant FTD and a mouse model carrying the mutant CHMP2B^Intron5^ transgene in forebrain neurons, the expression levels of miR-124 were markedly decreased in the cortex. Furthermore, the levels of targets of miR-124, most notably AMPA receptor subunits GluA2 and GluA4, were upregulated in the cortex of the mouse brain and the prefrontal cortex of FTD patients, leading to an enrichment of calcium-impermeable receptors, which are responsible for the age-dependent deficits in sociability in CHMP2B^Intron5^ mutant mice [93]. Studies profiling miRNAs in several brain regions of patients with Parkinson’s disease (PD) have revealed the downregulation of miR-34b/c in both the late (clinical) and early (pre-motor) stages of the disease. miR-34b/c downregulation is correlated with a reduction in DJ1 and parkin levels, two proteins associated with familial forms of PD, and results in compromised cell viability, which is accompanied by mitochondrial dysfunction and oxidative stress [94]. A review of Huntington’s disease (HD) showed that many miRNAs had altered expression levels in the brain; for instance, the downregulation of miR-124 and miR-132 was observed in the brain of an HD mouse model (R6/2) [95]. Improved performance in the rotarod test was observed with miR-124 and miR-132 overexpression [96,97]. These studies indicate that miRNAs can improve the molecular, pathological, and behavioral phenotypes observed in HD models.

#### 2.2.7. miRNA and Post-Traumatic Stress Disorder

PTSD is a neuropsychiatric disorder occurring in susceptible individuals following exposure to a traumatic event. Patients with PTSD are known to show enhanced fear conditioning and to benefit from exposure-based therapy, which is very similar to the fear extinction training used in animals. Therefore, to examine the role of miRNAs in PTSD-related symptoms in animal models, studies focus on their involvement in fear conditioning and fear extinction [98]. Many reports have shown miRNAs’ dysregulation in the context of stress for both preclinical and clinical models of PTSD [98,99], including the previously mentioned miR-128b [39], miR-182 [35], miR-34a [21], miR-92 [42], and miR-132 [30].

Increasing evidence suggests that miRNAs may play a prominent role in moderating or mediating the impact of severe stress and trauma susceptibility/resilience [98,100]. The potential of circulating miRNAs to be used as biomarkers of both vulnerability and resilience to stress was examined. Prior to the stressful event, four miRNAs (miR-24-2-5p, miR-27a-3p, miR-30e-5p, miR-362-3p) were significantly decreased only in rats that later became vulnerable to stress. After stress exposure, four different miRNAs (miR-28-3p, miR-99b-5p, miR-139-5p, miR-326-3p) were decreased in the resilient subgroup [101]. Recently, a mouse model of PTSD, the arousal-based individual screening model, was developed. Through this model, two subpopulations were identified, one consisting of susceptible mice showing long-lasting hyperarousal with persistent PTSD-like phenotypes, and the other of resilient mice showing normal arousal and behaving just like control mice [102]. By using this model, miRNAs targeting two key PTSD-related genes (*FKBP5* and *BDNF*) were evaluated, and lower transcript levels of miR-15a-5p, miR-497a-5p, and miR-511a-5p were found in the hippocampus and hypothalamus of susceptible mice compared to resilient mice [100]. Moreover, the expression of miR-Let-7e in mice’s prefrontal cortex has been demonstrated to differentiate restraint-stress-resilient genotypes from susceptible genotypes [103]. These findings indicate that PTSD susceptibility/resilience might be shaped by miRNAs.

## 3. Preclinical Application

Although it is still not clear whether alterations in miRNAs are a cause or consequence of neurodegeneration, or whether they are associated with memory impairments, the identification of altered circulating miRNAs in these diseases, including the fact that patients with AD are characterized by extracellular miRNA alterations in their body fluids, such as blood and CSF, has prompted the use of these non-coding sequences as biomarkers for the disease [90,104,105,106]. With only a few FDA-approved therapies, effective treatment strategies for AD are urgently required [107,108]. Currently, the diagnostic biomarkers recognized for AD are Aβ peptides and tau proteins detected in CSF and positron emission tomography scans; however, these are limited by their invasiveness and high cost. Therefore, blood-based AD biomarkers, such as miRNAs, are highly desirable to evaluate the efficiency of therapeutic agents in clinical trials and accelerate the therapeutic discovery process [109]. Another potential application of miRNAs is as predictors of progression from mild cognitive impairment (MCI) to AD. Currently, some miRNAs have been identified to have the potential to target genes directly involved in AD pathophysiology, as discussed above. Notably, members of the miR-132 (miR-128, miR-132, and miR-874) and miR-134 (miR-132, miR-323-3p, and miR-382) families have emerged as potential biomarkers for early dementia. Expression levels are dynamically altered in MCI and AD patients [74]. Data mining techniques have identified increased expression of miR-34a in the temporal cortex of AD patients, which correlates with the severity of AD pathology as well as with the increased expression of miR-34a in the 3xTg-AD mouse model [110]. Remarkably, miR-34c also belongs to a 6-member small non-coding RNA signature obtained from CSF exosomes of AD patients and can be used to predict the progression of MCI to AD [88]. Outside the CSF, the expression of miR-34a and miR-34c has been shown to be increased in the cellular and/or plasma constituents of blood specimens of AD patients compared with normal elderly controls [25,111]. Furthermore, the expression level of miR-34c has been inversely correlated with the extent of dementia, as measured by the MMSE, suggesting the potential use of this miRNA as a biomarker for disease progression [87]. A 5-year longitudinal follow-up study revealed that an increase in plasma levels of miR-206 may be used to predict cognitive decline and progression towards dementia at the MCI stage [92]. In addition, previous studies have documented the upregulation of miR-206 in the hippocampus, CSF, and plasma of animal models for AD, such as Tg2576 and APP/PS1 transgenic mice, as well as in the brains of patients with AD. Further, miR-206-dependent dysregulation of BDNF has been reported to contribute to the pathogenesis of AD [67,112].

Currently, studies on miRNAs as biomarkers show inconsistent results. However, some of them have a relatively high efficiency, specificity, and sensitivity for AD diagnosis. There are challenges and limitations, such as whether the temporal changes in miRNA profiles are AD-specific and are correlated between different sources, from patients’ brains to biofluids, and how to divide the boundaries when opposing changes occur during different disease stages or in different patient subsets [16]. It is also important to investigate the collective roles of multiple miRNAs when targeting the same etiological factor, in order to practice the idea of combining multiple miRNAs as biomarkers, and for a clear understanding of diagnosis, subset differentiation, prediction, and precision medicine in AD, together with the application of systems biology.

## 4. Conclusions and Outlook

The present review summarized the miRNAs participating in the regulation of learning and memory in vivo. Furthermore, we discussed the importance of functional and dysfunctional miRNAs in dementia, with an emphasis on AD. Although the signaling pathways and mechanisms of these individual miRNAs during the pathophysiological process have been elucidated, the overall effect of intricate miRNA networks will not be fully understood until global miRNA profiling and genome-wide studies on their targets in the brains of patients with AD have been intensively investigated.

The potential application of miRNAs as biomarkers and non-invasive diagnostic tools has made advances. Currently, over 500 miRNAs that have been used in biomarker-related studies are registered in NIH (clinicaltrials.gov), and certain platforms have been established for specific functions, including risk prediction and multistage disease progression for different diseases and conditions. However, there is still a lack of consistency among many miRNA signatures. For example, factors such as the sample sources, miRNA detection methods, cohorts, sample size, and patients with comorbidities may influence these molecules, making it more challenging to design efficient strategies [113]. miRNA-based clinical trials and applications in neurodegenerative diseases, particularly AD, are scarce. Targeted or site-specific drug delivery in specific brain regions with the use of non-invasive medical devices and the safety- and tolerability-relevant miRNA effects in clinical trials are key issues that should be addressed in the future.

## Figures and Tables

**Figure 1 biomedicines-10-01856-f001:**
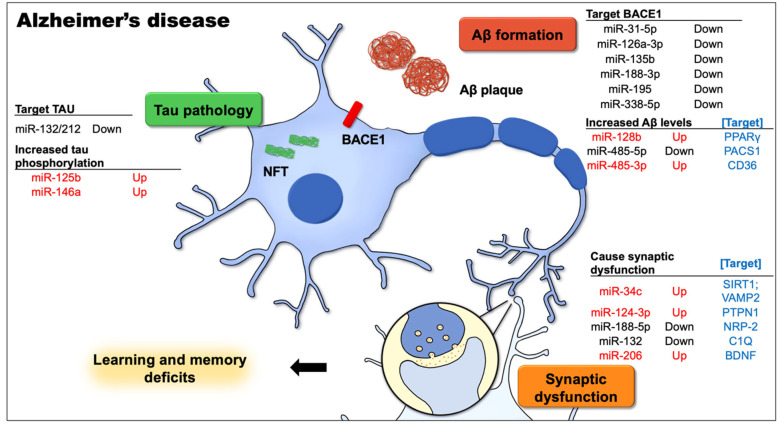
This figure demonstrates the most well-known Alzheimer’s disease (AD)-related cellular and signaling pathways in neurons that contribute to learning and memory deficits, and the dysregulated microRNAs along with the targets involved in the pathophysiological process of AD. The causative factors for AD, including the aggregation of intracellular amyloid-β (Aβ), expression levels and activity of β-site APP cleavage enzyme 1 (BACE1), extracellular amyloid plaques, and neurofibrillary tangles (NFTs) from hyperphosphorylated tau, together with Aβ-mediated synapse elimination and synaptic failure, may lead to cognitive impairment.

**Table 1 biomedicines-10-01856-t001:** Manipulation of miRNA levels in regulating learning and memory in wild-type rodents.

miRNA	Change Direction	Method	Fold Change	Location	Behavioral Tasks	Effects in Learning/Memory	Reference
miR-34a	Down	miRNA sponge	-	Basolateral amygdala	Auditory fear conditioning	Impaired	[21]
	Up	Viral overexpression	-	Lateral ventricle	MWM	Enhanced	[22]
	Up	Transgenic overepxression	Thousands	Whole brain	T-maze	Impaired	[23]
miR-34c	Up	Viral overexpression		Hippocampus	MWM	Impaired	[41]
miR-92	Down	miRNA sponge	-	Hippocampus	CFC	Impaired	[42]
miR-125b	Up	Injection of mimic	2	Hippocampus	CFC	Impaired	[43]
miR-126a-3p	Up	Viral overexpression	1.5~2	Hippocampus	CFC; MWM; NOR	Enhanced	[37]
miR-126a-3p	Down	Virus-mediated interference	0.4			Impaired	
miR-128b	Down	Virus-mediated interference	-	Infralimbic prefrontal cortex	Contextual fear extintion	Impaired	[39]
	Up	Viral overexpression	-			Enhanced	
miR-132-3p	Up	Transgenic overepxression	1.5	Hippocampus	Barnes maze	Enhanced	[33]
	Up		3		NOR; Barnes maze	Impaired	
	Up	Viral overexpression	2	Perirhinal cortex	NOR	Impaired	[32]
	Down	Virus-mediated interference		Hippocampus	Trace fear conditioning	Impaired	[30]
	Up	Injection of mimic	2	basolateral amygdala	Learned safety	Enhanced	[40]
	Down	Knock out (KO)	-			Impaired	
miR-132/212	Down	Knock out (KO)		excitatory forebrain	NOR; CFC; Barnes maze	Impaired	[29]
miR-134	Up	Viral overexpression	6	Hippocampus	CFC	Impaired	[28]
miR-137	Up	Viral overexpression	-	Hippocampus	CFC; MWM	Impaired	[44]
	Down	conditional KO	0.5	Hippocampus	MWM; Barnes maze	Impaired	[45]
miR-146a	Down	Virus-mediated interference	-	Hippocampus	CFC; NOR; Object location memory test	Impaired	[46]
miR-182	Up	Injection of mimic	4	Lateral amygdala	Auditory fear conditioning	Impaired	[35]
miR-183/96/182	Up	Viral overexpression	4~12	Hippocampus	NOR	Enhanced	[36]
	Down	miRNA sponge	0.5			Impaired	
miR-195	Down	Injection of antagomir	0.5	Hippocampus	MWM	Impaired	[47]
miR-335-5p	Up	Injection of mimic	-	Hippocampus	MWM	Impaired	[48]
miR-466f-3p	Up	Viral overexpression	1.5	Hippocampus	MWM; NOR; Barnes maze	Enhanced	[5]
	Down	miRNA sponge	-			Impaired	

CFC: Contextual fear conditioning; NOR: novel object recognition; MWM: Morris water maze.

**Table 2 biomedicines-10-01856-t002:** Manipulation of miRNA levels in regulating learning and memory in transgenic, mutant, or stressed rodents.

miRNA	Change Direction	Method	Fold Change	Location	Behavioral Tasks	Effects in Learning/Memory	Strains of Disease Model	Reference
miR-31-5p	Up	Viral overexpression	3	Hippocampus	T-maze; NOR; Barnes maze	Enhanced	3xTg-AD	[49]
miR-34c	Down	Injection of inhibitor	-		CFC	Rescued and enhanced	APP/PS1-21	[24]
	Down	Injection of inhibitor	0.5	Third ventricle	MWM	Rescued	SAMP8	[25]
miR-107	Down	Injection of mimic	2	Hippocampus	MWM	Rescued	Aβ ICV injection model	[50]
miR-124	Down	Injection of LNA-probe	0.4	Hippocampus	MWM	Rescued	EPAC−/−	[27]
	Up	Viral overexpression	-			Impaired	EPAC+/+	
	Up	Viral overexpression	-	Hippocampus	MWM	Impaired	Tg2576	[51]
	Down	Injection of antagomir	-			Rescued		
miR-124-3p	Up	Viral overexpression	5~10	Lateral ventricle	MWM	Rescued	APP/PS1	[52]
miR-126a-3p	Up	Viral overexpression	1.5~2	Hippocampus	CFC; MWM; NOR	Enhanced	APP/PS1	[37]
miR-128b	Down	Knock out (KO)		* Cerebral cortex	MWM	Rescued	3xTg-AD	[53]
miR-132-3p	Up	Viral overexpression	2.5	Lateral ventricle	MWM	Rescued	APP/PS1	[54]
	Up	Viral overexpression	1.5	Hippocampus	MWM	Rescued	Aβ ICV injection model	[55]
miR-132/212	Up	Injection of mimic	-	Ventricles	Barnes maze	Rescued	3xTg-AD	[56]
miR-134	Down	Injection of LNA-probe	0.6			Rescued	SIRT1 mutant	[28]
miR-135b	Up	Injection of mimic	5	Third ventricle	Y-maze	Rescued	SAMP8	[57]
miR-135b-5p	Up	Viral overexpression		basolateral amygdala	Acute restraint stress and auditory fear conditioning	Enhanced	Stress resilient mice	[38]
	Down	Injection of inhibitor				Rescued	Stress susceptible mice	
miR-139	Up	Injection of mimic	2	Hippocampus	CFC; MWM; NOR	Impaired	SAMP8	[58]
	Down	Injection of inhibitor	0.4			Rescued		
	Down	Injection of inhibitor	-	Hippocampus	Y-maze; MWM	Rescued	5xFAD	[59]
miR-181a	Up	Viral overexpression	4	Hippocampus	MWM	Rescued	APP/PS1	[60]
miR-188-5p	Up	Viral overexpression	-	Hippocampus	T-maze; CFC	Rescued	5xFAD	[61]
miR-188-3p	Up	Viral overexpression	-	Hippocampus	MWM	Rescued	5xFAD	[62]
miR-195	Up	Injection of mimic	2			Rescued	A chronic brain hypoperfusion model	[47]
	Up	Viral overexpression	-	Hippocampus	MWM	Rescued	APP/PS1	[63]
	Up	Viral overexpression	-	Hippocampus	NOR	Rescued	ApoE4KI +/- 5XFAD	[64]
miR-196a	Up	Viral overexpression	1.6	Hippocampus	MWM	Rescued	Aβ ICV injection model	[65]
miR-200b/c	Up	Injection of mimic	-	Lateral ventricle	Barnes maze	Rescued	Aβ ICV injection model	[66]
miR-206	Down	Injection of antagomir	-	Cerebral ventricles	CFC; Y-maze	Rescued	Tg2576	[67]
miR-338-5p	Up	Viral overexpression	2	Hippocampus	MWM	Rescued	5xFAD	[68]
	Up	Viral overexpression	4~5	Hippocampus	MWM	Rescued	APP/PS1	[69]
miR-485-5p	Up	Viral overexpression	4	Hippocampus	MWM; CFC	Rescued	APP/PS1	[70]
miR-485-3p	Down	Injection of inhibitor	-	Lateral ventricle	Y-maze	Rescued	5xFAD	[71]

CFC: Contextual fear conditioning; NOR: novel object recognition; MWM: Morris water maze. * Only this brain region has been used to performed experiments.

**Table 3 biomedicines-10-01856-t003:** Summary of deregulated miRNAs, target genes, and functions in AD models and locations in AD patients.

miRNAs	Reported Changes	Strains	Behavioral Task in Rodents	Target Genes	Function	Patient Samples (Brain)		References		
						Brain	Peripheral Body Fluid	Animal Models	Huamn Brain	Human Peripheral Body Fluid
miR-31-5p	Down	3xTg-AD	T-maze; NOR; Barnes maze	APP/BACE1	Reduces plaque load and intraneuronal Aβ		serum	[49]		[86]
miR-34c	Up	APP/PS1-21	CFC	SIRT1; VAMP2	Induces synaptic failure and memory deficits	Hippocampus	plasma; CSF	[24]	[24,82]	[25,87,88]
	Up	SAMP8	MWM	SYT1/ROS-JNK-p53 pathway	Mediates synaptic and memory deficits by targeting SYT1		* serum	[25]		[25]
miR-107	Down	Aβ ICV injection model	MWM	BDNF-TrkB and AKT pathways	Prevents Aβ-induced cell death, Aβ and Tau accumlation	Neocortex	plasma	[50]	[72]	[89]
miR-124-3p	Down	APP/PS1	MWM	C1QL3	Cerebromicrovascular impairments, breakdown of BBB and promote angiogenesis	Neocortex; hippocampus		[52]	[72,73,75]	
	Up	Tg2576	MWM	PTPN1	Impairs synaptic transmission by reducing AMPA trafficking	Temporal cortex; hippocampus		[51]	[51]	
miR-126a-3p	Down	APP/PS1	CFC; MWM; NOR	EFHD2/BACE1	Reduces Aβ plaque area and neuroinflammation	Temporal cortex		[37]	[72,74]	
miR-128b	Up	3×Tg-AD	MWM	PPARγ	Its knockout suppresses amyloid plaque formation, Aβ generation and neuroinflammation	Cerebral cortex	plasma	[53]	[53]	[80]
miR-132/212	Down	3XTg-ADKO	Barnes maze	TAU	Regulation of Aβ metabolism and tau pathology	Temporal cortex; prefrontal cortex; hippocampus	plasma	[56]	[56,74,75,76]	[81]
miR-132	Down	Aβ ICV injection model	MWM	MAPK1	Inhibits iNOS expression and oxidative stress			[55]		
	-	APP/PS1	MWM	C1Q	Induces synaptic proteins (PSD95, Synapsin-1, p-Synapsin) expression			[54]		
miR-135b	Down	SAMP8	Y-maze	BACE1	Promotes proliferation and neuroprotective effect	Prefrontal cortex	serum	[57]	[75]	[57]
miR-139	Up	SAMP8	CFC; MWM; NOR	CB2	Decreases responses to proinflammatory stimuli	Neocortex		[58]	[72]	
miR-146a	Up	5xFAD	Y-maze; MWM	ROCK1/PTEN signal pathway	Induces tau hyperphosphorylation	Temporal lobe neocortex; hippocampus	serum	[59]	[72,77]	[86]
miR-181a	Down	APP/PS1	MWM	FOXO1	Ameliorates amyloid plaque deposition, decelerates pericyte apoptosis and BBB breakdown	Neocortex	* plasma	[60]	[72]	[90]
miR-188-5p	Down	5xFAD	CFC; T-maze	NRP-2	Reduces Aβ-mediated reduction in dendritic spine density and basal synaptic transmission	Cortex; hippocampus		[61]	[61]	
miR-188-3p	Down	5xFAD	MWM	BACE1	Reduces Aβ formation and suppresses neuroinflammation	temporal lobe		[62]	[62]	
miR-195	Down	ApoE4KI +/− 5XFAD	NOR	ApoE-synj1-PIP2 pathway	Rescues lysosomal defects in neurons and cognitive deficits in mice	Parietal cortex	* CSF	[64]	[64]	[64]
	Down	APP/PS1	MWM		Prevents Aβ Deposition by Inhibiting APP Expression			[63]		
miR-196a	Down	Aβ ICV injection model	MWM	LRIG3	Stimulates neuronal survival and inhibits its apoptosis through represses oxidative stress and inflammation			[65]		
miR-206	Up	Tg2576	CFC; Y-maze	BDNF	Reduces dendritic spine density	Temporal cortex	* serum/plasma	[67]	[67]	[91,92]
miR-338-5p	Down	5xFAD	MWM	BACE1	Decreases Aβ formation and reduces neuroinflammation	Hippocampus		[68]	[68]	
		APP/PS1	MWM	BCL2L11	Attenuates amyloid plaque deposition, attenuates neuron apoptosis			[69]		
miR-485-5p	Down	APP/PS1	MWM; CFC	PACS1	Represses Aβ40-induced pericyte apoptosis	Prefrontal cortex		[70]	[75]	
miR-485-3p	Up	5xFAD	Y-maze	CD36	Induces Aβ plaque accumulation, tau pathology development, neuroinflammation, and cognitive decline	Frontal cortex; precentral gyrus	CSF; plasma	[71]	[71]	[71]

BBB: blood–brain barrier; CFC: contextual fear conditioning; NOR: novel object recognition; MWM: Morris water maze; AD: Alzheimer’s disease; CSF: cerebrospinal fluid. * Samples from patients with mild cognitive impairment (MCI).

## Data Availability

Not applicable.

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
