# Peer review of "MicroRNAs in Learning and Memory and Their Impact on Alzheimer’s Disease"

_biomedicines, 2022, doi:10.3390/biomedicines10081856_

Round 1

Reviewer 1 Report

Review article titled (MicroRNAs in learning and memory and their impact on Alzheimer’s disease) by Wang et al. discussed the role of miRNAs in the process of learning and memory and their dysfunction in Alzheimer's disease.

1- Line 79: correct it to "Role of..."

2- Table 1 & 2 must be included within the paper NOT as supplementary files

3- Table 1 should be divided into smaller 2 or 3 tables based on a selected criterion

4- Introduction: intense focus on fearing activity is confusing as it is not the target.

5-line 143: 2.1.2. Regulation of inhibitory learning by miRNAs

this title is not adequate and needs rewriting

6-Introduction should gives an account on AD and main symptoms and known pathologies at least beta amyloid and tau proteins as in the last part of the review.

7-Cocnlusion: is in some parts overstated such as "The use of miRNAs as biomarkers"; this is still not confirmed an needs a metanalysis or more studies to be confirmed or approved. 

Author Response

Thanks for your comments and suggestions, most of which are constructive and very helpful.
We have submitted  the point by point response and the manuscript with revisions. Please see the attachment.

Reviewer 2 Report

The manuscript by Wang et al entitled "MicroRNAs in learning and memory and their impact on Alzheimer’s disease" is relatively well-written and scientifically sound. The only drawback is the language, some improvements can be made, e.g., "The formation of new memories is a complex process that requires activity-dependent gene transcription, a dynamic expression profile of newly synthesized proteins, and finely tuned specific neuronal networks, such as memory traces/engrams to strengthen particular synaptic connections and sustain neuronal plasticity [6]":. The sentence is too long and gets somewhat incomprehensible, there are many more such sentences in the manuscript.

Author Response

(The authors gave the same response as above.)

Reviewer 3 Report

In this review, Wang et al. discuss the role of MicroRNAs in learning and memory and their impact on Alzheimer’s disease. The manuscript is well-written and very interesting. I have only one minor comment that may improve this review:

-          The Authors might want to add and discuss other studies in which have been reported a key role for miRNA in stress-related disorder. I may suggest to add, for example, a study in which as been described a possible role for miR-15a-5p, miR-497a-5p and miR-511-5p in traumatic stress susceptibility/resilience (PMID: 34068160). In particular, a brain region-dependent dysregulation of these miRNAs has been detected in traumatic stress susceptible mice, which showed long-lasting social/cognitive deficits in a previous work (PMID: 33392367). This is just an example, the Authors might want to add other studies covering the link between stress/trauma and memory/learning.

Author Response

(The authors gave the same response as above.)

Round 2

Reviewer 1 Report

The revised version is adequately preparaed